# ECAvatar: 3D Avatar Facial Animation with Controllable Identity and Emotion

## ABSTRACT

Speech-driven 3D facial animation has attracted considerable attention due to its extensive applicability across diverse domains. The majority of existing 3D facial animation methods ignore the avatar's expression, while emotion-controllable methods struggle with specifying the avatar's identity and portraying various emotional intensities, resulting in a lack of naturalness and realism in the animation. To address this issue, we first present an *Emolib* dataset containing 10,736 expression images with eight emotion categories, i.e., neutral, happy, angry, sad, fear, surprise, disgust, and contempt, where each image is accompanied by a corresponding emotion label and a 3D model with expression. Additionally, we present a novel 3D facial animation framework that operates with unpaired training data. This framework produces emotional facial animations aligned with the input face image, effectively conveying diverse emotional expressions and intensities. Our framework initially generates lip-synchronized and expression models separately. These models are then combined using a fusion network to generate face models that effectively synchronize with speech while conveying emotions. Moreover, the mouth structure is incorporated to create a comprehensive face model. This model is then fed into our skin-realistic renderer, resulting in a highly realistic animation. Experimental results demonstrate that our approach outperforms state-of-the-art 3D facial animation methods in terms of realism and emotional expressiveness while also maintaining precise lip synchronization.

## CCS CONCEPTS

• **Computing methodologies** → **Animation**; **Computer graphics**.

## KEYWORDS

Virtual Avatar, Speech-driven, Facial Animation, Emotional Controllable

## 1 INTRODUCTION

Speech-driven facial animation is a long-standing problem in computer vision and computer graphics. In contrast to 2D facial animation [9, 16, 17, 29, 33, 38], 3D facial animation methods [7, 8, 10, 28, 40] offer a unique advantage in creating facial animations with

*ACM MM, 2024, Melbourne, Australia*

© 2024 Copyright held by the owner/author(s). Publication rights licensed to ACM.
ACM ISBN 978-x-xxxx-xxxx-x/YY/MM
https://doi.org/10.1145/nnnnnnn.nnnnnnn

diverse poses, viewpoints, and lighting conditions, making them focal points for both academic and industrial communities.

Due to the strong correlation between speech and lip movements, existing 3D animation generation methods mainly focus on lip synchronization, which may result in a less natural or even completely static upper face. To address this issue, some studies incorporated blinking or head movements to enhance the naturalness of generated animations [31, 41]. However, these approaches fail to consider the emotional expression of the avatar and cannot effectively respond to the emotion conveyed in the speech clip, which is unfavorable for generating 3D face animations with a high degree of realism and naturalness. In recent years, some researchers have concentrated on and made significant strides in the emotion-controllable methods via specifying emotion labels [8, 27]. However, due to the limitation of the training set (e.g., MEAD [38] or RAVDESS [21]), the identity diversity of the generated animations still remains to be improved. In addition, to the best of our knowledge, the current methods mainly focus on face models without considering the oral cavity, leading to deficiencies in the realism of the generated animations.

In this paper, we propose an *Emolib* dataset consisting of 10,736 expression images in eight categories: neutral, happy, angry, sad, fear, surprise, disgust, and contempt. Each image is associated with a label pair, which includes the emotion category and intensity, along with a corresponding 3D face model with expression. Moreover, to address the issue of insufficiently conveying emotion in previous methods, we propose *ECAvatar*, a speech-driven framework that produces 3D facial animations corresponding to the emotions in the speech. The framework takes a face image and a speech clip as inputs. The face image is used to reconstruct a 3D neutral expression face model. And the speech clip is processed by a speech emotion recognition module to generate a label pair that indicates both the emotion category and emotional intensity. Then, based on the label pair, an emotional face model that closely resembles the neutral face model is retrieved in the *Emolib* dataset. Meanwhile, a lip-synchronized model with mouth structure is generated with the speech clip and the neutral face model. A fusion network *FuNet* is proposed to incorporate the emotional and lip-synchronized model to obtain a comprehensive face model, which is then fed into a skin-realistic renderer to generate expressive animation. We evaluate our framework on three databases and compare it with nine representative methods. Extensive experimental results show that our framework achieves enhanced realism while maintaining satisfactory lip synchronization.

We summarize our contributions as follows:

(1) We construct an *Emolib* dataset of 10,736 emotional face models with diverse identities, comprising eight emotional categories with three intensities. Each item contains a face image with a label pair (emotion category and intensity), and its corresponding 3D face model with expression.

(2) We propose *ECAvatar*, a novel framework capable of generating 3D facial animation using only unpaired training data. This framework allows users to easily define the avatar's identity, accurately express various emotions based on input speech, and ensure seamless lip synchronization.

(3) Our framework not only generates complete head models with the internal mouth structure, but also incorporates a skin-realistic renderer for more photo-realistic face animations.

## 2 RELATED WORK

### 2.1 Emotionless 3D face animation

Compared to early approaches, which tend to specify the mapping rules between speech and facial motions explicitly [35, 36, 42], deep learning methods prefer to use large amounts of data to implicitly learn the relationship. VOCA [7] is a speaker-independent method that can capture a wide range of speaking styles but fails to synthesize the upper face movements. MeshTalk [31] focuses on the upper part of the face, which is lacking in VOCA. Greenwood et al. [11] mainly leverages BLSTM to consider the facial expression and head pose with respect to the input speech. Richard et al. [30] then proposes a fusion model to combine lip and eye movements together. Although all of the above works achieved good results, none of them considered complete facial motion. FaceFormer [10] encodes the long-term audio context and autoregressively predicts a sequence of animated 3D face meshes based on transformer [37]. CodeTalker [40] models the generation as a code query task in a finite proxy space of the learned codebook to promote vividness. Recently, some approaches have also been developed based on the diffusion model. FaceDiffuser [34] is a non-deterministic deep learning model to generate speech-driven facial animations that is trained with both 3D vertex and blendshape-based datasets. DiffSpeaker [23] is a Transformer-based network equipped with novel biased conditional attention modules that steer the attention mechanisms to concentrate on both the relevant task-specific and diffusion-related conditions. However, these methods ignore the effect of speech emotions on expressions, resulting in less natural animations.

### 2.2 Emotional 3D face animation

It is well observed that when spoken with different emotions, even the same sentence often elicits distinct facial expressions. Consequently, an increasing number of researchers recognize the importance of introducing emotion for facial animation synthesis. Karras et al. [18] designs an end-to-end convolutional network that employs linear prediction coding to encode audio and then maps the speech data to vertex coordinates of a 3D face model. Additionally, the network uses an emotion vector latent code as the additional input to control speaking styles, facial expressions, and emotional states. Pham et al. [28] trains an LSTM-RNN neural network on a large-scale audiovisual dataset to achieve a time-varying contextual non-linear mapping between audio streams and facial movements with implicit emotional awareness. 3D-TalkEmo[39] adds expression to neutral 3D meshes by a multi-dimensional scaling-based projection method to generate emotional 3D face animation. Speech4Mesh[12] utilizes speech information to reconstruct 3D

data and encode emotions as embedding vectors to control emotional states. EmoTalk [27] introduces the emotion disentangling encoder to disentangle the emotion and content in the speech and then employs an emotion-guided fusion decoder to generate a 3D talking face with enhanced emotion. However, the training data used in EmoTalk requires the manual labor of several professional animators. EMOTE [8] employs a content-emotion exchange mechanism to supervise different emotions on the same audio, but users need to manually specify emotion labels. Moreover, the characteristics of the oral cavity and human skin are disregarded, making the generated animation less realistic. Therefore, we propose an emotional facial animation framework to overcome these problems.

## 3 PRELIMINARIES

### 3.1 3D face model representation

Inspired by recent work on speech-driven facial animation [7, 10, 40], we use the FLAME model [20] as the face representation, which allows for intuitive control and editing of facial shape, pose, and expressions using a few parameters. It includes identity-specific face shape parameters $\beta \in \mathbb{R}^{|\beta|}$, expression parameters $\psi \in \mathbb{R}^{|\psi|}$, and pose parameters $\theta \in \mathbb{R}^{3k+3}$ ($k = 4$, for the left and right eyeballs, neck, and jaw, respectively), which can be defined as:

$$M(\beta, \theta, \psi) \rightarrow \{V, F\}, \tag{1}$$

where $V \in \mathbb{R}^{N \times 3}$ is the set of vertices and $F \in \mathbb{Z}^{+M \times 3}$ is the set of faces formed by the index of $V$. $N = 5023$ and $M = 9976$ denote the number of vertices and faces, respectively.

### 3.2 3D face model reconstruction

EMICA [8] is utilized to reconstruct 3D face models with expression from images. It introduces a depth perceptual emotional consistency loss to ensure that the reconstructed results are consistent with the expressions depicted in the input images. In addition to the mesh model, the FLAME parameters for shape, expression, and jaw pose will also be provided. However, the 3D face models reconstructed by EMICA still retain the pose and position corresponding to the input image. We desire the lip-synchronized and the emotional face model that needs to be fused to remain in the same pose; this ensures that regardless of the emotional models used, the resulting comprehensive face model from the fusion network maintains a consistent pose, which will enhance the natural transition between different emotions. Therefore, a model alignment operation is necessary. In this work, we apply the Iterative Closest Point (ICP) algorithm [3], which is widely used for achieving the optimal rigid transformation between two meshes, to obtain a FLAME model with pose parameters $\theta = 0$.

### 3.3 Emolib dataset

To address the limited identity diversity in existing affective 3D datasets, we introduce the *Emolib* dataset. We select a subset of the AffectNet dataset [24] and generate 3D models corresponding to the images in this subset to form the *Emolib* dataset used in our work.

The AffectNet dataset comprises an extensive collection of images depicting facial expressions. The eight distinct emotion categories represented in these manually annotated images include

**Table 1: The number of images for each emotion category.**

| Emotion | Neutral | Happy | Sad | Surprise | Fear | Disgust | Anger | Contempt |
|---|---|---|---|---|---|---|---|---|
| Level1 | 500 | 500 | 500 | 262 | 108 | 205 | 500 | 378 |
| Level2 | 500 | 500 | 500 | 500 | 500 | 500 | 500 | 500 |
| Level3 | 500 | 500 | 359 | 500 | 500 | 424 | 500 | 500 |

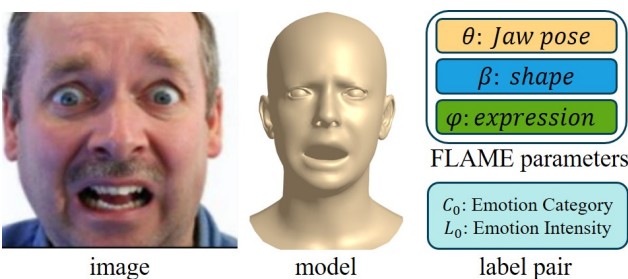

image · model · label pair

$\theta$: Jaw pose
$\beta$: shape
$\varphi$: expression

FLAME parameters

$C_0$: Emotion Category
$L_0$: Emotion Intensity

**Figure 1: An example item of *Emolib*. It contains a facial image, a corresponding 3D model, a label pair, and Flame parameters.**

neutral, happy, angry, sad, fear, surprise, disgust, and contempt. Furthermore, the dataset includes labels denoting valence and arousal that are associated with every image. Arousal relates to the intensity of an emotion or the power of the related emotional state, whereas emotional valence specifies whether an emotion is positive or negative [6].

For each category, the largest arousal value, $V_{aromax}$, and the lowest arousal value, $V_{aromin}$, are identified. Starting from $V_{aromin}$, the arousal values in each category were divided into three levels with an interval of $(V_{aromax} - V_{aromin})/3$. Each picture is classified into one of the three levels according to its arousal value. 500 pictures are randomly selected in each level (if the total number of pictures in the level is less than 500, all of them are included). The number of images included in each category is shown in Table 1. Subsequently, the 3D face model reconstruction (Section 3.2) is employed to generate the corresponding 3D mesh model of each image as well as the corresponding FLAME parameters. After these steps, we obtain the dataset *Emolib*. It contains facial images with expressions that are divided into eight categories, and each category is further divided into three levels of intensity. Each image corresponds to(Figure 1): (1) a label pair, which consists of emotion category and intensity; and (2) a 3D face model, which includes a mesh model and FLAME parameters for shape, expression, and jaw pose.

### 3.4 VOCATeeth dataset

We incorporate the mouth structure into all 123,341 models in the VOCASET [7], forming the VOCATeeth dataset. It is utilized to generate lip-synchronized models incorporating the mouth structure, hence improving the realism of face animation.

The teeth and tongue models are generated using FaceGen SDK[1], and subsequently, the upper and lower teeth models are manually separated (Figure 2(a)above). We translate, scale, and rotate the teeth

[1]https://facegen.com/sdk.htm

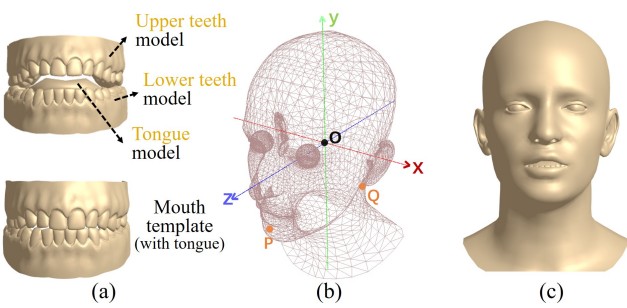

**Figure 2: The mouth structure. (a) The mouth structure includes teeth and tongue models. (b) The coordinate system in the face model. (c) A complete face model with the mouth structure.**

and tongue models (Figure 2(a)below), generating a mouth template to align with the FLAME topology. VOCASET comprises 12 subjects and offers corresponding 12 face templates, each featuring a neutral expression and closed mouth. Due to the non-rigid deformation of the lips when talking, it is challenging to obtain a reasonable teeth motion trajectory based on the vertex displacements of the lip region. However, according to our observation, we discovered that compared to the lips, the motion of the chin is more consistent with the trajectory of the teeth. Thus, for each subject, we compute the chin motion of each face model relative to the corresponding template and migrated the motion to the mouth template, forming the mouth model corresponding to that face model. Then, we merge them to generate a complete head model containing the mouth structure. The computational details are as follows:

For each subject, we first set up a coordinate system in the corresponding face template (Figure 2(b)): the origin $o$ is at the center of the face model, the line from right ear to left ear forms the x-axis and determines the $+x$ direction, the head is oriented in the $+y$ direction, and the face is oriented in the $+z$ direction. A point $P_n$ in the chin region and a point $Q_n$ below left ear are selected and projected onto the $yoz$ plane to obtain $P'_n$ and $Q'_n$, respectively, and then connected to obtain the line segment $P'_n Q'_n$. Since all face models have the same topology, we traverse the face models belonging to this subject other than the template, performing the following steps: The corresponding points $P_t$ and $Q_t$ are selected, and projected onto the $yoz$ plane to obtain $P'_t$ and $Q'_t$, respectively, and connected to obtain the line segment $P'_t Q'_t$. The angle $\theta$ between the two segments is calculated. The mouth model $M_{mouth}$ is obtained by keeping the mouth template's upper teeth model stationary and rotating the lower teeth and the tongue model around a selected point in the $yoz$ plane, by the same angle $\theta$. Subsequently, we integrate it with the face model and obtain the complete head model(Figure 2(c)). The complete head model consists of 15,051 vertices and 29,780 faces, with 10,028 vertices and 19,804 faces from the added mouth structure.

We iterate through the 12 subjects to ensure that all VOCASET models incorporate mouth structures, which ultimately constitute the VOCATeeth dataset.

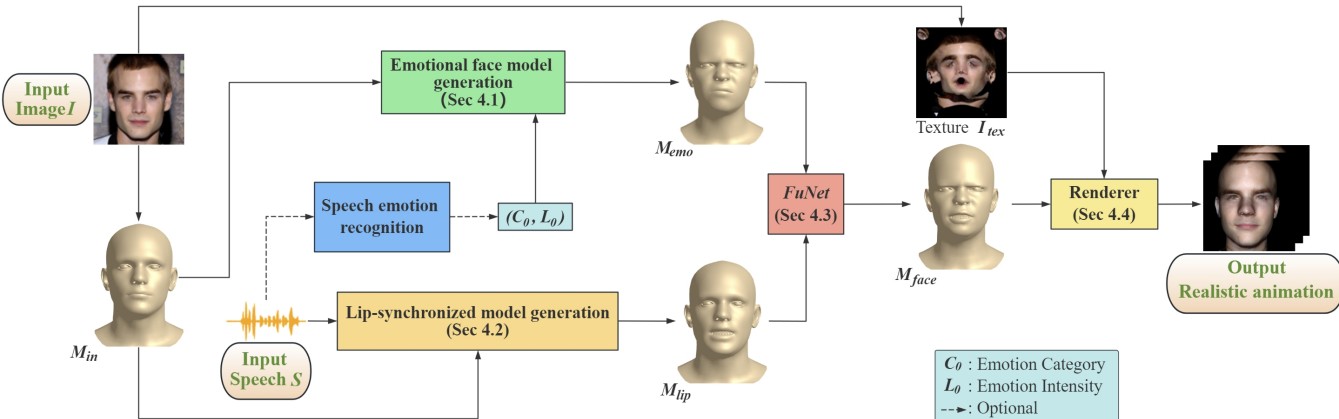

**Figure 3: The pipeline of our method. A face image and a speech clip are used to generate models with and without expressions, which are fused with a *FuNet* network to obtain a complete face model. The output is fed into a skin-realistic renderer to generate the final animation.**

## 4 METHOD

Given a face image and a speech clip as inputs, we aim to generate 3D facial animations corresponding to the emotions in the speech clip. The framework of our method is illustrated in Figure 3. Given a face image $I$, we first reconstruct the corresponding 3D face model and remove the emotion to obtain a neutral face model $M_{in}$. Then, we recognize the emotion of the input speech $S$ and retrieve an emotional face model $M_{emo}$ that matches both the identity and emotion from the *Emolib* database (Section 4.1). Subsequently, we generate a lip-synchronized model $M_{lip}$ that contains a mouth structure from $S$ and $M_{in}$ (Section 4.2). After that, the fusion network *FuNet* generates a fused model $M_{face}$ with both emotion and synchronized lips from $M_{emo}$ and $M_{lip}$ (Section 4.3). Finally, the $M_{face}$ model undergoes the skin-realistic renderer to apply texture and form video frames (Section 4.4).

**Generation of the neutral face model corresponding to input image.** To provide identity information for the subsequent lip-synchronized and face model generation, we generate a neutral expression model $M_{in}$ based on the input image $I$. We first reconstruct the face model corresponding to the input image using the method mentioned in Section 3.2. Then, we set its expression parameters to zero, and convert it to a mesh model, thus obtaining a neutral face model $M_{in}$ without expression.

### 4.1 Emotional face model generation

To generate an emotional face model, we first retrieve the model $M_{emori}$ in the *Emolib* (Section 3.3) that is most similar to the neutral model $M_{in}$ based on the emotion label pair $(C_0, L_0)$ ($C_0$ represents the category of the emotion and $L_0$ represents the intensity of the emotion). The emotion label can either be predicted by speech emotion recognition or provided by the user.

For automatic emotion label prediction, we employ the speech emotion recognition framework [22]. This framework predicts nine different types of emotions, and we exclude two emotion categories that are not available in *Emolib*, choosing only the remaining seven as output. Its output *EmoOut* is in the form : *EmoOut* =

$\{P_{neutral}, P_{happy}, P_{angry}, P_{sad}, P_{disgusted}, P_{fearful}, P_{surprised}\}$. The values indicate the probabilities that the input speech emotion is neutral, happy, angry, sad, disgusted, fearful or surprised, respectively. The category with the largest value is adopted as the $C_0$ in the label pair. As previous research [5] identifies the trend that the intensity of an emotion is positively related to its probability, we design a linear mapping method to obtain the corresponding emotion intensity $L_0$ from the speech recognition output. Subsequently, we select the subset in *Emolib* corresponding to the intensity $L_0$ in emotion category $C_0$. In this subset, our method retrieves the most similar model $M_{emori}$ for $M_{in}$ based on the Euclidean distance between their FLAME shape parameters.

To address fluctuations in both emotion categories and intensity within speech over time, we segment the speech into short clips lasting $t$ seconds. We then analyze each segment to identify the predominant emotions, enabling us to retrieve the corresponding emotional model $M_{emori}$. To ensure smooth transitions between different emotion categories and intensities, we establish a transition period $t_{trans}$. During the initial and final $t_{trans}$ seconds of each $t$-second window, we interpolate between the emotional models of the current and adjacent windows using the formula $M_{emo} = (1 - j/fn) \cdot M_{emori1} + (j/fn) \cdot M_{emori2}, j = \{1, 2, \cdots, fn\}$. Here, $fn$ represents the number of frames in the transition period, and $M_{emori1}$ and $M_{emori2}$ are the models for adjacent segments. During the middle segment of each time window, $M_{emo}$ remains fixed at $M_{emori}$. This principle extends to the initial $t_{trans}$ seconds of the first window and the final $t_{trans}$ seconds of the last window, where interpolation is unnecessary.

### 4.2 Lip-synchronized model generation

To generate a lip-synchronized model, our framework employs a modified SelfTalk model [26]. The input of SelfTalk includes identity information and audio features, which are extracted by wav2vec2 [1] from the input speech $S$. Through experimentation, we noticed that using HuBert [13] for audio feature extraction yields superior results. Thus, we substitute the usage of wav2vec2

with HuBert. To augment the realism of the facial animation with teeth and tongue, we incorporate the mouth structure into the lip-synchronized model by retraining the modified SelfTalk model using the VOCATeeth dataset (section 3.4). The input and output dimensions of the modified SelfTalk are adjusted to $(15051 \times 3)$ to fit the VOCATeeth dataset. The model $M_{in}$ is incorporated with mouth structure and, together with the input speech $S$, is fed into the modified and retrained SelfTalk model, to generate a lip-synchronized model sequence.

## 4.3 Fusion network

We propose a fusion network *FuNet* to fuse the emotional face model $M_{emo}$ (Section 4.1) and the lip-synchronized face model $M_{lip}$ (Section 4.2) to generate anemotional lip-synchronzed face model $M_{face}$.

**Architecture**. The *FuNet* architecture (Figure 4) comprises four convolutional layers, three residual blocks, and two fully connected layers, with detailed parameter specifications provided in Table 2. It should be clarified that the input to *FuNet* consists of two face models, each comprising 5,023 vertices. The mouth region of the model generated by *FuNet* is expected to be similar to that of $M_{lip}$, which is determined by the design of the loss functions. To improve fusion efficiency, we remove the mouth structure from $M_{lip}$ before being used as input. Furthermore, the output of *FuNet* does not include the mouth structure, and will be further merged with the teeth and tongue models in the $M_{lip}$ as the final output $M_{face}$. The initial two convolutional layers are dedicated to extracting features from the input data. The residual blocks accelerate the training speed and facilitate better feature propagation. Subsequently, the latter two convolutional layers refine the extracted features, thereby augmenting the representation capability of the network. Finally, the fully connected layers are applied to aggregate all features before normalizing them to meet the dimension requirements of FLAME models.

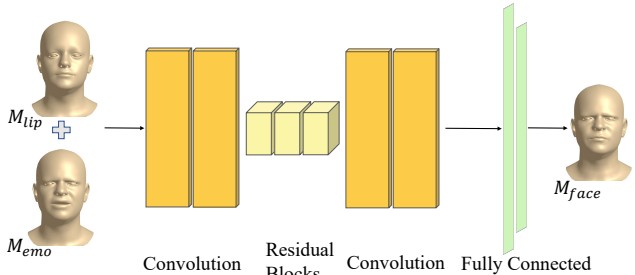

**Figure 4: The architecture of *FuNet*. It includes four convolutional layers, three residual blocks, and two fully connected layers.**

**Loss function**. There are four terms in loss functions of *FuNet*. According to the description and annotations provided by FLAME, the 3D face model could be segmented into different regions, including the chin region $C$, the lip region $\mathcal{L}$, the other region inside the face $\mathcal{F}$, and regions outside the face $O$ (Figure 5 for more details). In the following, $v_{lip}$ are vertices of $M_{lip}$, $v_{emo}$ are vertices of $M_{emo}$ and $v_{out}$ are vertices of the fused model $M_{face}$.

**Table 2: The *FuNet* parameters. The first parameter in the output denotes the number of convolutional kernels.**

| Type | Kernel | Stride | Output | Activation |
|---|---|---|---|---|
| Input | - | - | $1 \times 5023 \times 6$ | - |
| Convolution | $3 \times 3$ | $1 \times 1$ | $256 \times 5023 \times 6$ | LeakyReLU |
| Convolution | $3 \times 3$ | $1 \times 1$ | $128 \times 5023 \times 6$ | LeakyReLU |
| Residual Blocks | - | - | $128 \times 5023 \times 6$ | LeakyReLU |
| Residual Blocks | - | - | $128 \times 5023 \times 6$ | LeakyReLU |
| Residual Blocks | - | - | $128 \times 5023 \times 6$ | LeakyReLU |
| Convolution | $3 \times 3$ | $1 \times 1$ | $64 \times 5023 \times 6$ | LeakyReLU |
| Convolution | $3 \times 3$ | $1 \times 1$ | $1 \times 5023 \times 6$ | LeakyReLU |
| Fully connected | - | - | 8192 | Linear |
| Fully connected | - | - | $5023 \times 3$ | Linear |

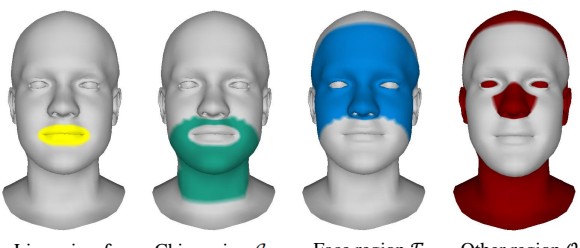

Lip region $\mathcal{L}$    Chin region $C$    Face region $\mathcal{F}$    Other region $O$

**Figure 5: Different regions of the 3D face model.**

(1) The lip-synchronized loss $L_{lip}$ is defined for maintaining an accurate correspondence between lip and speech:

$$L_{lip} = \|v_{out} - v_{lip}\|_2^2, v_{out}, v_{lip} \in \mathcal{L}. \tag{2}$$

(2) The chin loss $L_{chi}$ is defined to make the mouth region driven by both speech and expression:

$$L_{chi} = w_1 * \|v_{out} - v_{lip}\|_2^2 + \\ w_2 * \|v_{out} - v_{emo}\|_2^2, \tag{3}$$

where $v_{out}$, $v_{lip}$ and $v_{emo}$ indicate the vertices that belong to the chin region $C$. $w_1$ and $w_2$ are weights set by users.

(3) The expression loss $L_{exp}$ is defined to ensure the fused model contains the specified emotion:

$$L_{exp} = \|v_{out} - v_{emo}\|_2^2, v_{out}, v_{emo} \in \mathcal{F}. \tag{4}$$

(4) The identity loss $L_{id}$ is defined for maintaining identity:

$$L_{id} = \|v_{out} - v_{lip}\|_2^2, v_{out}, v_{lip} \in O. \tag{5}$$

The overall loss function is defined as follows:

$$L = L_{lip} + L_{chi} + L_{exp} + L_{id}. \tag{6}$$

**Training**. The training set consists of 81,223 model pairs. Each pair includes a model with expression and a model without expression. The models with and without expressions are randomly paired. The models without expressions are sourced from the VOCASET dataset; they are used as the $M_{lip}$ input for *FuNet*. In addition, we select 81,223 images (not included in *Emolib*) under categories other than neutral in the AffectNet dataset and reconstruct these images to obtain the corresponding 3D models with expressions, which are used as the $M_{emo}$ input for *FuNet*. Each 3D face model is represented as a tensor of $5023 \times 3$. When feeding the data into the

                                    Anonymous Authors

neural network, it is essential to horizontally concatenate two 3D face models to form a tensor of size $5023 \times 6$. The parameters in the chin loss are $w_1 = 0.35$ and $w_2 = 0.65$. We train the *FuNet* model on a single NVIDIA GeForce RTX 3090 for 750 epochs with the Adam optimizer ($\beta_1 = 0.9, \beta_2 = 0.999$), with learning rate $lr = 0.0001$ and batch size $bs = 16$.

## 4.4 Skin renderer

In this section, we develop a skin-realistic renderer based on the realistic skin rendering model [25] to improve the realism of the output animation. The inputs to the renderer are the face model $M_{face}$ and a texture map $I_{tex}$ reconstructed from the input image $I$ by the FLAME texture expansion[2].

The rendering equation is defined as:

$$L_o(\mathbf{x}_o, \omega_o) = \sum_A \sum_{\Omega^+} S(\mathbf{x}_o, \omega_o, \mathbf{x}_i, \omega_i) L_i(\mathbf{x}_i, \omega_i) \cos\theta_i \Delta\omega_i \Delta A_i, \quad (7)$$

where the variables $x_o$ and $\omega_o$ denote the exiting point and the outgoing direction, respectively. Similarly, $x_i$ and $\omega_i$ represent the incident point and incident direction. $\Omega^+$ is the hemisphere determined by the surface normal and contains all possible incident light directions $\omega_i$. $\theta_i$ is the angle between the incident light direction $\omega_i$ and the surface normal direction. $L_i$ and $A$ denote the radiant illumination and the surface of the object, respectively.

In this work, we use the BSSRDF equation proposed by Jensen [14] to represent the term $S(\mathbf{x}_o, \omega_o, \mathbf{x}_i, \omega_i)$, which contains the specular reflection term $S_r(\omega_o, \omega_i)$ as well as the diffuse reflection term $S_d(\mathbf{x}_o, \omega_o, \mathbf{x}_i, \omega_i)$:

$$S(\mathbf{x}_o, \omega_o, \mathbf{x}_i, \omega_i) = S_r(\omega_o, \omega_i) + S_d(\mathbf{x}_o, \omega_o, \mathbf{x}_i, \omega_i). \quad (8)$$

A modified Kelemen/Szirmary-Kalos BRDF [19] is used to simulate the specular reflection term $S_r$:

$$S_r(\omega_o, \omega_i) = \frac{D(\omega_o, \omega_i, \mathbf{n}_o, \alpha) F(\omega_o, \omega_i)}{h \cdot h}, \quad (9)$$

where the Fresnel-Schlick equation [32] is used to approximate the Fresnel equation $F(\omega_o, \omega_i)$. And $h$ denotes the half-angle vector between the incident light direction and view direction. The Beckmann normal distribution function [2] is used to calculate the ratio of the microfacets that are oriented in the same direction as the half-angle vector $D(\omega_o, \omega_i, \mathbf{n}_o, \alpha)$.

For highly scattering materials like skin, multiple scattering dominates. Therefore, the proposed realistic skin renderer neglects the influence of single scattering, and the term $S_d$ is defined as:

$$S_d(\cdot) = \frac{1}{\pi} F_t(\mathbf{x}_o, \omega_o) R_d(r) F_t(\mathbf{x}_i, \omega_i), \quad (10)$$

where $F_t$ is the Fresnel transmittance and $R_d(r)$ denotes the diffusion profile [15].

# 5 EXPERIMENT

## 5.1 Evaluation metrics

We follow previous works [10, 27, 31] to compute the Lip Vertex Error (LVE) to measure lip synchronization. This metric computes the average $L_2$ error of the lip region vertices. Given that our framework generates animations with emotional expressions, solely measuring

---

[2]https://github.com/TimoBolkart/TF_FLAME/blob/master/fit_2D_landmarks.py

---

the lip region is insufficient. Therefore, we incorporate the Emotional Vertex Error (EVE) proposed in EmoTalk [27] to assess the maximum $L_2$ error of the vertex coordinate displacement in the eye and forehead regions.

## 5.2 Datasets

Three datasets, IEMOCAP [4], RAVDESS [21], and VOCASET, were employed to evaluate our framework.

The VOCASET dataset contains about 29 minutes of 4D scans captured at 60 fps and synchronized audio. The dataset has 12 subjects and 480 sequences of about 3-4 seconds. The sentences are selected from a set of standardized protocols to maximize speech diversity. Since the 3D face model provided in VOCASET does not contain expressions, we only calculated the **LVE** results on it.

The IEMOCAP dataset includes 302 recorded conversation videos. Each segment is annotated to determine whether it consists of one or more emotions in nine distinct categories. Additionally, the attributes of valence, arousal, and dominance are included. As this dataset lacks videos of real people speaking alongside corresponding audio, we used the results generated from this dataset only for the **user study**.

The RAVDESS dataset includes recordings from 24 professional actors. They read two sentences with different emotions: neutral, happy, sad, angry, fear, surprised, and disgusted. Each expression has two levels of emotional intensity. We performed a frame-by-frame reconstruction of the video and calculated the **EVE** results using the obtained 3D expression model as ground truth. The results generated on this dataset were also used for the **user study**.

## 5.3 Comparison with state-of-the-art methods

We compared our framework with nine state-of-the-art speech-driven 3D avatar animation generation methods: VOCA [7], Face-Former [10], MeshTalk [31], CodeTalker [40], SelfTalk [26], FaceDiffuser [34], DiffSpeaker [23], EmoTalk [27] and EMOTE [8]. As far as we know, EmoTalk currently only supports several specific identities, and MeshTalk doesn't employ FLAME model, making them difficult to generate facial animations with the identities in the RAVDESS and VOCASET datasets, so we only compare with them in the qualitative evaluation.

**Qualitative evaluation.** In Figure 6, we show comparisons of results when the intensity of eight categories of emotions (neutral, happy, angry, sad, disgust, fear, surprise, and contempt) is maximized (level=3). The first column is the user input image, and the second column is the emotion label pair. It can be observed that VOCA, FaceFormer, CodeTalker, SelfTalk, FaceDiffuser and DiffSpeaker fail to reflect the emotions. Although MeshTalk incorporates subtle facial expressions like frowning and blinking, it still struggles to discern the emotional states of the characters from their facial expressions. EmoTalk, EMOTE and our method can generate obvious expressions. However, we note that the mouths of EMOTE and EmoTalk avatars tend to be wide open in order to present obvious expressions, which may significantly affect lip synchronization. In addition, the EMOTE avatar lacked mouth structures, and EmoTalk only filled the cavity by supplementing a few faces between the upper and lower lips, both of which could not correctly reflect the

**Figure 6: The comparison results of state-of-the-art methods and ours. The face models in the same row are generated with the same speech clip. It can be observed that our method has good lip synchronization while expressing speech emotion.**

avatar's teeth movements when speaking, affecting the sense of realism. Nevertheless, our approach achieves a good balance between lip synchronization and emotional performance.

**Quantitative evaluation.** We measured lip synchronization by calculating the LVE on the test set of the VOCASET. As shown in Table 3, our framework achieves the best lip synchronization, better than other methods. The utilization of a modified SelfTalk model in generating the lip-synchronized model leads to a closer resemblance to SelfTalk in the results, but better performance is achieved by the replacement of the speech feature extraction module. In future endeavors, the performance of LVE can be enhanced further by using a superior model, and this flexibility is also an advantage of our framework. EMOTE has the maximal LVE value, which may be due to the fact that the mouths of avatars tend to be wide open to present obvious expressions, resulting in unsatisfactory lip synchronization.

The comparison of EVE was conducted solely on the RAVDESS database due to the absence of facial expressions in the VOCASET models and the unavailability of video in the IEMOCAP for reconstructing face models. The ground-truth models were reconstructed frame-by-frame from videos in RAVDESS. To verify the generalization performance of all the methods, we used their pretrained models and test on the RAVDESS dataset. Evidently, our method outperforms the other methods in terms of performance on EVE. The disparity among emotionless methods is negligible, as their models only produce neutral expressions, which are inconsistent with the ground truth with emotion. It is observed that EMOTE exhibits a higher EVE value, potentially attributed to its ability to

**Table 3: LVE and EVE evaluation results. Our method outperforms other methods on both LVE and EVE.**

| Method | LVE($\times 10^{-5}$) ↓ | EVE($\times 10^{-5}$) ↓ |
|---|---|---|
| VOCA | 4.05 | 2.71 |
| FaceFormer | 3.81 | 2.73 |
| CodeTalker | 3.47 | 2.77 |
| SelfTalk | 2.88 | 2.92 |
| FaceDiffuser | 4.24 | 2.69 |
| DiffSpeaker | 3.32 | 2.77 |
| EMOTE | 6.46 | 3.13 |
| Ours | **2.74** | **1.34** |

generate highly obvious expressions. In contrast, the emotional expression of the ground truth is less apparent, leading to a greater disparity from the ground truth, which even surpasses the difference between neutral expressions and the ground truth. Our model demonstrates superior generalization performance, possibly attributed to the enhanced identity diversity of our approach in comparison to EMOTE.

The aforementioned results demonstrate that our framework achieves better results in terms of emotional expression while maintaining good lip synchronization, thereby enhancing realism with accurate preservation of speaking contents.

**User study.** 26 participants were recruited to evaluate the animation quality, including 13 males and 13 females ranging in age from

**Table 4: The user study results and the percentage indicate that our method is better than comparison methods.**

| Ours vs. Competitor | Realism ↑ | | | | | | Lip Sync ↑ |
|---|---|---|---|---|---|---|---|
| | Sad | Disgust | Fear | Angry | Happy | Surprise | |
| Ours vs. VOCA | 76.92% | 80.77% | 69.23% | 69.23% | 76.92% | 69.23% | 57.69% |
| Ours vs. FaceFormer | 46.15% | 50.00% | 69.23% | 57.69% | 46.15% | 57.69% | 53.85% |
| Ours vs. MeshTalk | 73.08% | 73.08% | 84.62% | 73.08% | 61.54% | 80.77% | 55.77% |
| Ours vs. CodeTalker | 53.85% | 50.00% | 65.38% | 73.08% | 61.54% | 69.23% | 44.23% |
| Ours vs. SelfTalk | 65.38% | 53.85% | 46.15% | 80.77% | 50.00% | 46.15% | 46.15% |
| Ours vs. FaceDiffuser | 50.00% | 73.08% | 61.54% | 53.84% | 80.77% | 69.23% | 61.54% |
| Ours vs. DiffSpeaker | 57.69% | 65.38% | 80.77% | 69.23% | 42.31% | 73.08% | 42.31% |
| Ours vs. EMOTE | 76.92% | 65.38% | 84.62% | 61.54% | 42.31% | 84.62% | 84.62% |
| Ours vs. EmoTalk | 80.77% | 38.46% | 73.08% | 53.85% | 50.00% | 84.62% | 71.16% |

18 to 36. We used speech clips from the RAVDESS and IEMOCAP datasets. Video pairs were presented randomly to participants, who were asked to choose the better video in terms of realism and lip synchronization. Each video pair contains a video generated by our method and a video generated by other methods. The results are shown in Table 4, which indicates the percentage of participants who chose our method's output over the other.

In terms of lip synchronization, our method achieved a notable advantage over methods with emotions, including EMOTE and EmoTalk. Compared to emotionless methods, we not only outperform other methods in quantitative results, but also have better or comparable results for human perception. In terms of realism, for the seven emotionless methods, our method has better results in almost all emotion categories. This suggests that our proposed scheme with emotion and mouth structure is effective in improving the realism of the animation. For methods with emotions, our method has also achieved better results in most emotion categories as well. This may be due to the addition of mouth structure. EmoTalk performs well under the disgust category, which may be due to the fact that this type of expression involves more movement around the lips, and we have made a small concession in the mouth movement in order to maintain better lip synchronization. In addition, we presented participants with the animation both with and without the mouth structure, and all participants unanimously agreed that the inclusion of the mouth structure enhanced animation realism.

**Table 5: The results of the ablation study.**

| | w/o $L_{lip}$ | w/o $L_{chi}$ | w/o $L_{exp}$ | w/o $L_{id}$ | Ours |
|---|---|---|---|---|---|
| LVE($\times 10^{-5}$) ↓ | 748.48 | 2.78 | 2.70 | **2.66** | 2.74 |
| EVE($\times 10^{-5}$) ↓ | 1.35 | 1.49 | 894.32 | 1.47 | **1.34** |

## 5.4 Ablation study

We conducted an ablation study on the loss functions, as shown in Table 5. The absence of $L_{lip}$ leads to a significant increase in LVE, suggesting that this loss is critical for lip synchronization. Although removing $L_{exp}$ and $L_{id}$ would improve the LVE, this term is also necessary since removing it would lead to an increase in EVE. The lack of $L_{chi}$, $L_{exp}$, and $L_{id}$ greatly worsens EVE because

the regions they affect are within the regions evaluated for EVE. We further conducted an ablation study on the choice of speech feature extractors. Hubert yielded an LVE of $2.74 \times 10^{-5}$, outperforming wav2vec's LVE of $2.89 \times 10^{-5}$. Thus, Hubert was selected for our framework. The results of the ablation experiments demonstrate the indispensability of every loss function within the *FuNet*.

## 6 LIMITATION AND FUTURE WORK

The expression models in the framework are retrieved from a database, which yields a higher diversity of expressions compared to directly reconstructing a dataset such as MEAD to obtain 3D training data. Although we assume that when the database is large enough, a model can be retrieved that accurately matches the identity information of the input images, it is still possible that the retrieved models do not represent the identity information very well. Currently we incorporate identity information in the lip-synchronized model and choose very similar identities in the emotional face model as well to solve this problem. In future work, we will enhance the importance of the identity information in the fusion network as well. In addition, there is a slight imbalance in the number of images within the intensity levels under some emotion categories due to the biased data distribution of AffectNet, which we will subsequently address by optimizing the division approach or expanding the dataset.

## 7 CONCLUSION

Accurately conveying emotions is essential for improving the realism of facial animations. In this paper, we present an *Emolib* dataset containing 10,736 expression images in eight categories with their corresponding emotion category and intensity labels, as well as 3D models with expressions. We also propose *ECAvatar*, a realistic 3D emotional facial animation framework, in which we introduce a skin-realistic renderer to obtain highly realistic facial animations that include mouth structures. Our framework, which solely relies on unpaired training data, enables users to easily define the avatar's identity. Moreover, it automatically adjusts to present various emotion categories and intensities based on input speech. Experimental results show that compared with SOTA 3D facial animation methods, our approach yields more realistic animations (e.g., good emotional performance) and preserves satisfactory lip synchronization.

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
