# OpenReview forum: "ECAvatar: 3D Avatar Facial Animation with Controllable Identity and Emotion"
_acmmm.org/ACMMM/2024/Conference — MM2024 Poster_

### Official Review · Reviewer_xziW · 2024-05-12

**Rating:** 5
**Confidence:** 3

**Summary:**

This paper introduces a novel framework for speech-driven 3D facial animation that can generate avatars with both controllable identity and emotion. The authors present an Emolib dataset containing a variety of facial expressions with corresponding emotion labels and 3D models. The framework operates on unpaired training data and is capable of producing emotional facial animations that are aligned with the input face image and speech clip. It initially generates separate lip-synchronized and expression models, which are then combined using a fusion network to create a comprehensive face model. This model is subsequently fed into a skin-realistic renderer to produce highly realistic animations.

**Strengths:**

1. A new large dataset of 10736 emotional face is proposed. Creatively adding mouth structure for better animation.

2. The proposed ECAvatar framework is innovative in its approach to generating 3D facial animations from unpaired training data, offering a high degree of control over the avatar's identity and emotion.

3. Using UV texture for talking head animation is a good attempt and has a clear future application.

4. Experiments conducted on diverse datasets using different methods have demonstrated consistent and promising results for the proposed approach.

**Limitations:**

1. The detailed analysis of the proposed dataset should be provided to prove its quality.

2. The texture extraction method is relatively old and coupled with light. Maybe some newer method can be considered for better looking.

3. The design of sub modules in section 4.1 and 4.2 is not clear enough to demonstrate innovative design.

4. If the dataset cannot be open source, the contribution to the community will be small.

**Suitability:**

3

---

### Official Review · Reviewer_qQhK · 2024-05-12

**Rating:** 4
**Confidence:** 3

**Summary:**

This paper first introduces a new dataset named Emolib, in which they design several strategies for generating annotation. Specifically, they selected a subset from AffectNet and annotated it with eight emotion labels and 3DMM parameters. They also incorporate mouth structure into VOCASET, making it the first 3D facial dataset that contains teeth and tongue annotations. Based on the above dataset, they propose a 3D facial animation framework that conducts emotional audio-driven facial animation. They also propose a skin-realistic renderer which improves the realism of the final animation. The experiment results show they can generate audio-lip synced facial videos under different emotions.

**Strengths:**

1. Propose a new dataset Emolib that is annotated with 3D face model and eight emotion categories with diverse identities. They incorporate mouth structure into VOCASET, making it the first 3D facial dataset that contains teeth and tongue annotations.
2.  Propose an audio-driven face animation framework, which can generate audio-lip syncretized face animations with emotion. They also propose a skin-realistic renderer which improves the realism of the final animation.
3. The paper is well-organized and easy to follow.

**Limitations:**

1. The mouth region in the result of the fusion network sorely comes from M_lip branch (line 485-490), which only encodes audio-synced features but not emotion-related information. As we know, emotion is related to the mouth, \eg, happiness leads to a smiling mouth. How do you guarantee the emotion of mouth region?
2. The supplementary video shows that the mouth differs with different emotions. How does this work? Does the emotion of the mouth region come from the neutral face template (line 384-321, due to the inferior decoupling ability in 3DMM)? Can you show some examples of neutral faces when the input image contains a high intensity of emotion?
3. The rendered results seem to be less audio-lip syncretized in the supp video.
4. How long does this method take to animate an avatar with 30s audio?
5. Since the proposed dataset contributes greatly to the final results, will you release your dataset?

**Suitability:**

3

---

### Official Review · Reviewer_vJJq · 2024-05-24

**Rating:** 5
**Confidence:** 3

**Summary:**

The paper presents ECAvatar, a novel framework for 3D facial animation that can synchronize lip movements with speech and express various emotions. The authors introduce an Emolib dataset and a skin-realistic renderer to enhance the animations' realism. The framework operates with unpaired training data and allows for customizing the avatar's identity and emotion based on input speech.

**Strengths:**

The Emolib dataset, consisting of diverse emotional expressions with corresponding 3D models, is valuable to the field. The introduction of a skin-realistic renderer and the ability to control identity and emotion in facial animations are notable strengths. The framework's capability to work with unpaired data is advantageous, reducing the need for extensive paired datasets. Will this Emolib dataset be open sourced in the future?

**Limitations:**

While the Emolib dataset offers diversity, the method's reliance on retrieving models from a database could limit the representation of certain identities. The mention of a slight imbalance in the dataset's intensity levels could impact the model's performance uniformly across all emotions. The paper could benefit from a more detailed comparative analysis with other state-of-the-art methods, particularly regarding computational efficiency and scalability. While the user study is positive, further evaluation on the generalization of the framework to other datasets would strengthen the paper.

**Suitability:**

3

---

### Official Review · Reviewer_8zLi · 2024-05-26

**Rating:** 3
**Confidence:** 2

**Summary:**

Existing speech-driven 3D facial animation methods lack naturalness and realism because they ignore the avatar's expression, and emotionally controllable methods have difficulty specifying the avatar's identity and depicting various emotional intensities. To address these issues, this paper collects a face dataset, Emolib, that contains a wide range of expressions and intensities, and proposes a 3D facial animation framework, ECAvator.This paper conducted experiments on three datasets, and the experimental results presented illustrate that ECAVator outperforms state-of-the-art 3D facial animation methods in terms of realism and emotional expression, while also maintaining precise lip synchronization.

**Strengths:**

1. The motivation of the paper is good, and the flow of the proposed method is reasonable.
2. Compared to other vertex-based methods, this paper reconstructs the teeth and renders the face texture.
3. The proposed method demonstrates excellent performance on three datasets.
4. The expressive face model is retrieved from the Emolib database, which is significantly different compared to other directly generated methods.

**Limitations:**

1. This paper uses a skin renderer to enhance the realism of the output model, but the paper does not quantitatively analyze the image quality or video quality.
2. In the quantitative experiment (Table 3), it would be better to give the experimental results of LVE and EVE under each expression.
3. No video is provided.

**Suitability:**

2

---

### Meta-Review · Area_Chair_2GQ8 · 2024-07-01

**Recommendation:** Accept (Poster)
**Confidence:** 4

**Metareview:**

Overall, authors have clearly stated their motivation and proposal, allowing the reviewers to understand the paper better. Furthermore, the authors have explained their methodology and presented their results well. In addition, authors have agreed to release their dataset, which is a big contribution to the community. Finally, authors have addressed the issues stated by reviewers with sufficient explanation and adjustment.